# MFIDMA: A Multiple Information Integration Model for the Prediction of Drug–miRNA Associations

**DOI:** 10.3390/biology12010041

**Published:** 2022-12-26

**Authors:** Yong-Jian Guan, Chang-Qing Yu, Yan Qiao, Li-Ping Li, Zhu-Hong You, Zhong-Hao Ren, Yue-Chao Li, Jie Pan

**Affiliations:** 1School of Electronic Information, Xijing University, Xi’an 710129, China; 2College of Agriculture and Forestry, Longdong University, Qingyang 745000, China; 3School of Computer Science, Northwestern Polytechnical University, Xi’an 710129, China; 4Key Laboratory of Resources Biology and Biotechnology in Western China, Ministry of Education, College of Life Science, Northwest University, Xi’an 710129, China

**Keywords:** drug–miRNA association, SDNE, Word2vec, SMILES, deep neural network, convolution neural network

## Abstract

**Simple Summary:**

Predicting the possible associations between drugs and miRNAs would provide new perspectives on miRNA therapeutics research and drug discovery. However, considering the time investment and expensive cost of wet experiments, there is an urgent need for a computational approach that would allow researchers to identify potential associations between drugs and miRNAs for further research. In this paper, we present a computational method in this field named MFIDMA for simplifying the screening process. We also collect high-quality datasets from the current database. We conduct experiments on the collected datasets to prove the excellent performance of the proposed model. The MFIDMA is intended to be useful for the prediction of associations between drugs and miRNAs, and to be effective for the development and research of miRNA-targeted drugs.

**Abstract:**

Abnormal microRNA (miRNA) functions play significant roles in various pathological processes. Thus, predicting drug–miRNA associations (DMA) may hold great promise for identifying the potential targets of drugs. However, discovering the associations between drugs and miRNAs through wet experiments is time-consuming and laborious. Therefore, it is significant to develop computational prediction methods to improve the efficiency of identifying DMA on a large scale. In this paper, a multiple features integration model (MFIDMA) is proposed to predict drug–miRNA association. Specifically, we first formulated known DMA as a bipartite graph and utilized structural deep network embedding (SDNE) to learn the topological features from the graph. Second, the Word2vec algorithm was utilized to construct the attribute features of the miRNAs and drugs. Third, two kinds of features were entered into the convolution neural network (CNN) and deep neural network (DNN) to integrate features and predict potential target miRNAs for the drugs. To evaluate the MFIDMA model, it was implemented on three different datasets under a five-fold cross-validation and achieved average AUCs of 0.9407, 0.9444 and 0.8919. In addition, the MFIDMA model showed reliable results in the case studies of Verapamil and hsa-let-7c-5p, confirming that the proposed model can also predict DMA in real-world situations. The model was effective in analyzing the neighbors and topological features of the drug–miRNA network by SDNE. The experimental results indicated that the MFIDMA is an accurate and robust model for predicting potential DMA, which is significant for miRNA therapeutics research and drug discovery.

## 1. Introduction

As the demand for medical care increases, the cost of drug development is growing and unacceptable [1]. The main reason for the relatively low productivity in the pharmaceutical industry is attributed to the high cost of searching for new drug targets. However, finding appropriate drug targets from the numerous and disorderly informatics is one of the important purposes of bioinformatics. For a long period, many studies on therapeutic targets have focused on protein and have spent much time and effort exploring the drug response of proteins. However, about 80% of approved drugs target protein and 99% of them target only specific proteins [2]. This means that there are still vast proteins that are “undruggable”. Therefore, some researchers have shifted their focus in target selection to other biological entities such as microRNA (miRNA).

MicroRNA is a kind of endogenous non-coding RNA with a length of about 20 nucleotides, existing in humans, plants, animals and viruses [3]. To date, about 2600 human mature miRNAs have been discovered [4]. A considerable amount of literature has been published on miRNAs regarding their biogenesis, mechanic of action and function [5,6,7]. Research in this area has shown that the abnormal expression of miRNAs is involved in plenty of diseases including cancer, neurologic disorders, autoimmune diseases and cardiovascular diseases [8,9,10,11,12]. Furthermore, from post-transcriptional regulation, miRNAs can affect the gene to produce specific proteins, including the aforementioned “undruggable” proteins. Thus, miRNAs are considered to be potential high-value therapeutic targets and identifying the underlying drug–miRNA associations has major implications for the pharmaceutical industry [13,14].

Many researchers believe that miRNA pharmacogenomics would promote the development of personalized medicine [15,16]. However, there are two main challenges for miRNA-target therapeutics: the effective means of delivering the therapeutic agents to the target tissues and the safety evaluation of the potential drug response [17]. In the first challenge, the problem of poor cell permeability and pharmacokinetics can be solved by Lipinski’s Rule of Five [18]. In the other challenge, it is inevitable to study the situation of the association between drugs and miRNAs. For most drugs, it is relatively difficult to completely identify their association with different miRNA profiling through wet experiments because it is an intricate problem concerning a series of factors, and it is also labor-intensive and time-consuming work. [19,20]. Even though much effort has been invested in identifying DMA by wet experiment, the existing knowledge about drugs and miRNAs is not sufficient for guiding miRNA-targeted drug research. For improving the research and development of miRNA-target therapeutics, we need to accelerate the identification of DMA for future research. Compared with wet experiments, the computational method is the better choice for completing this mission, since it is lower in cost and higher in efficiency [21]. In particular, machine learning has made great contributions in the field of bioinformatics [22,23,24,25]. In molecular biology research, novel datasets and innovated concepts are being generated [26,27,28,29]. Thus, it is important to adopt techniques that can handle these data efficiently. Machine learning can process the vast amount of data generated by new high-throughput devices to extract undiscovered relationships that exist and are imperceptible to experts [30,31,32,33].

After years of efforts, several computational methods for predicting DMA have emerged. One category of these methods was based on the self-similarity network and the association network. For example, Lv et al. developed a model based on the drug–miRNA network to identify DMA. They constructed the drug–miRNA integrated network and applied a random walk with restart (RWR) algorithm to predict the underlying miRNA targets of drugs. This model can predict related miRNAs for drugs in the absence of known drug–miRNA associations, but it is sophisticated and contains too many adjusted parameters [34]. Furthermore, Qu et al. presented an in silico method for DMA prediction called HSDMA, which was also based on the drug–miRNA similarity network. [35]. They introduced the path-based relevance measurement method of HeteSim. In the HeteSim method, considering different search paths between the miRNAs and drugs is the most predominant issue, because the path in the heterogeneous network has semantics [36]. It can predict potential DMA by calculating the association score of each drug–miRNA pair based on the given search path, but the function for integrating different patterns of the search path is relatively simple. Moreover, Guan et al. proposed a prediction model called GIDMA. Inspired by the concept of graphlet interactions, they defined 28 types of graphlet interaction isomers that contained 1 to 4 vertexes and various connection patterns for describing the different relationships between 2 nodes [37]. Thereafter, the association score between the drugs and miRNAs was calculated based on the number of each isomer on the self-similarity network [38]. Furthermore, Wang et al. designed an DMA prediction model called RFDMA. This model combined the integrated similarity of miRNA and the drugs, and predicted DMA using the random forest algorithm [39]. Qu et al. presented a new method called TLHNDMA based on a triple-layer heterogeneous network. This network not only used data on drug self-similarity and miRNA self-similarity but also considered disease similarity. An iterative updating algorithm was also developed to propagate information in the network and complete the prediction task [40]. Additionally, Zhan et al. proposed a model called SNMFDMA, which did not directly use the similarity matrixes of drugs and miRNAs. They first used symmetric non-negative matrix factorization to process the similarity matrixes to generate new similarity matrixes. The Kronecker product of the new similarities matrixes was then regarded as the similarity of the drugs and miRNAs. Finally, regularized least squares were implemented to predict the potential associations between drugs and miRNAs [41].

Another category of prediction methods leverages other features to represent the drugs and miRNAs instead of self-similarity. An example of this is the study carried out by Huang et al. in which they constructed an end-to-end model named GCMDR to discover associations between miRNA and drug resistance. These authors combined the side information such as the miRNA expression profile, drug substructure fingerprints, gene ontology and disease ontology as attribute features of the miRNAs and drugs. This model used GCN to learn low-dimensional embedding vectors for each biological entity and predicted the association between the drugs and miRNAs [42]. Yu et al. built a web server for predicting the effects of drugs on miRNAs. They utilize k-mer, sequence information and the MACCS fingerprints to represent the miRNAs and drugs. The regulation of the miRNA expression of the drugs was then predicted using random forests [43].

In our paper, we propose a novel multiple features integration method based on the integration of multiple features, named MFIDMA. First, a bipartite network was established to represent the relationship between drugs and miRNAs. Second, the structural deep network embedding (SDNE) algorithm was implemented to extract topology information and generate the embedding vectors of each node in the network. Third, the miRNAs were directly represented using sequences and the drugs were indicated by simplified molecular input line entry specification (SMILES). The Word2vec algorithm was then adopted to extract attribute features. Finally, two kinds of features were separately entered into the convolutional neural network (CNN) and the deep neural network (DNN) for deep learning feature extraction and classification. Figure 1 provides the flowchart of the MFIDMA model.

In experiments, the known drug–miRNA pairs were collected from three databases including ncDR [44], RNAInter [45] and SM2miR [46]. It is worth noting that the SM2miR database was divided into three datasets according to its versions. After preprocessing these databases, there were three datasets available: ncDR, RNAInter and SM2miR. For evaluating the prediction ability of the MFIDMA, we implemented the proposed model on those three datasets and obtained average accuracies of 86.46%, 87.56% and 82.16% under a five-fold cross-validation. The average AUC values achieved 0.9407, 0.9444 and 0.8919 on ncDR, RNAInter and SM2miR, respectively. In addition, serval experiments were conducted for performance comparisons with respect to the choices of features and prediction methods. Furthermore, we carried out case studies using hsa-let-7c-5p and Verapamil to prove the prediction ability of the proposed method. There are 9 of the top 15 predicted drugs and 10 of the top 15 predicted miRNAs confirmed by the PubMed database, respectively. The results of the cross-validations and case studies demonstrated that the MFIDMA model could predict DMA accurately and robustly. This study may be helpful for predicting drug response and overcoming drug resistance for subsequent treatment and improving the situation for drug-target discovery.

## 2. Materials and Methods

### 2.1. Dataset

In previous studies, a large number of drug–miRNA interaction associations have been accumulated. We collected datasets from three databases including RNAInter, ncDR and SM2miR. Before preprocessing, we collected a total of 19,310 miRNA drug interaction samples from the ncDR, RNAInterer and SM2miR database websites. For clarity, there are three versions of SM2miR because they were updated on 10 June 2012, 28 August 2013 and 27 April 2015. To distinguish them from different versions, we named them SM2miR v1.0, SM2miR v2.0 and SM2miR v3.0. We also adopted the latest SM2miR v3.0 and refer to it as SM2miR in this paper. To improve our work, we only selected associations that related to the “*Homo sapiens*” type in the three datasets. By doing this, we collected a total of 12,323 DMA as the positive dataset, which included 470 different types of drugs and 1623 different types of miRNAs. Then, we constructed the negative dataset by randomly selecting the same number of negative samples as the positive samples from the unlabeled data. The distribution of the individual datasets is illustrated in Table 1. These positive samples can be represented as an adjacent edge list and then turned into a drug–miRNA association bipartite graph. The miRNA name and sequence recording in miRBase represent the information from each miRNA node. Similarly, the drug information is uniquely identified using the CID and SMILES from PubChem.

The PubChem database is a comprehensive substance and compound database, including data sources and contents and data organization. It not only provides the chemical structures and properties but also provides pharmacology and biochemistry information [47,48,49]. In the database, the chemical structure of the drug is represented by SMILES, which is an extensively used chemical notation system. It can encode chemical molecules through ASCII codes and is extensively used in chemical computer applications [50,51]. We collected a total of 492 different drugs and their corresponding SMILES from the PubChem database. 

The MiRBase database is a central online repository for nomenclature and sequences of miRNAs [52,53,54]. We obtained the sequences of 1788 miRNAs from the miRBase database. All miRNA sequences were identified on miRBase. 

### 2.2. Representation of miRNAs and Drugs with Word Embedding

Deep learning is currently the focus of machine learning in the field of computer vision and natural language processing. One reason for the sharp rise in the use of deep learning algorithms is because these algorithms are a powerful method for processing gigantic amounts of unsupervised data for downstream tasks [55]. The sequences of biomolecules and the structure of chemical compounds are intrinsic properties of miRNAs and drugs. Inspired by Buchan et al., miRNA sequences and drug SMILES could be presumed as “sentences”, while nucleotides and atoms are naturally “words” [56]. Therefore, the DMA datasets can be the text corpus for the learning representation vectors by Word2vec. The Word2vec model is a famous machine learning technique in the text processing field in recent years. It is a kind of distributed representation method and aims to connect different dimensions by coding [57]. If the words are similar in the context, the representation vectors are similar, either semantically or grammatically. Word2vec contains two important models, Skip-gram and CBOW. In this study, the CBOW model is implemented to generate embedding vectors by predicting the central word according to context. Instead of the traditional neural net language model, the model is constructed using an input layer, an output layer and projection layers. The framework of CBOW is illustrated in Figure 2. As shown in Figure 2, the vocabulary size is denoted as V and the size of the projection layer is represented as N. In the input layer, $v_{t-2}$, $v_{t-1}$, $v_{t+1}$ and $v_{t+2}$ represent the context of $v_{t}\;$and initial words are expressed as one-hot codes. The weights matrix between the input layer and the projection layer is represented by a V×N matrix M. The $M^{\prime }$ is not the transpose of M, but a N×V weights matrix between the projection layer and the output layer. The projection vectors $v_{p}$ are obtained using the weighted average of word vectors of context through the projection layer, following:(1)$$v_{p}=\frac{1}{4}M\left(v_{t-2}^{T}+v_{t-1}^{T}+v_{t+1}^{T}+v_{t+2}^{T}\right)$$
where M represents the weight matrix, $v_{c-2}$, $v_{c-1}$, $v_{c+1}$, $v_{c+2}$ represents context one-hot vectors of the c-th central word and $v_{p}$ represents the output of the projection layer. In the output layer, the probabilities that denote the appropriate center word are calculated through weight matrix $W^{\prime }$ and projection vectors. To predict an appropriate center word by minimizing the loss function:(2)$$\begin{array}{ll} E & =-\log\sum_{c=1}^{T}P\left(v_{t}|v_{t-2},v_{t-1},v_{t+1},v_{t+2}\right)\\ & {}_{}=-\log\sum_{t=1}^{T}\frac{\exp[u_{t}^{T}(v_{t-2}+v_{t-1}+v_{t+1}+v_{t+2})]}{\sum_{j\in V}\exp[u_{j}^{T}(v_{t-2}+v_{t-1}+v_{t+1}+v_{t+2})]} \end{array}$$
where $u_{c}$ represents the c-th row of weight matrix $W^{\prime }$. In this paper, we utilized the Word2vec algorithm to learn a fixed-length vector for representing the sequences. The Word2vec algorithm is implemented on a Python package named Gensim. Gensim Word2vec is practical for transforming each letter in the sequences into a vector. It is applied to process drug SMILES and miRNA sequences in this study. We set the parameters “vector size” to 64 and “minimum step size” to 1 for containing all of the letters in the sequences. Other parameters are default. Thereafter, each letter in the sequences will be represented as a vector with dimension 64.

### 2.3. Representing the Association between Drugs and miRNA with Graph Embedding

Network-based features are well proven to perform well in the link prediction tasks of heterogeneous graphs [58,59]. The topological feature represents the global structure of the bipartite graph. In contrast to previous studies, which extracted topology information from the network degree and clustering coefficient, the network embedding methods learn low-dimensional representations of nodes in the network [60]. To gain the highly non-linear structure from the bipartite graph, the graph embedding model SDNE is applied to formulate topological features [61]. The deep neural network in SDNE is more effective than shallow models to capture non-linear structures in the network. The SDNE has good performance in sparse networks since it combines first-order and second-order proximity for preserving the structure information in the network. Structural deep network embedding is an expansion of LINE [62], in which the definition of first-order and second-order proximity is identical to LINE. In the framework of SDNE, an unsupervised autoencoder is designed to extract the global structure of the network by preserving the second-order proximity. The similarities of pairwise nodes in the network are defined as the first-order proximities. A supervised component according to the Laplace matrix is designed to mine the information in the latent space by the first-order proximity. Finally, SDNE utilizes the deep autoencoder with multiple non-linear layers to represent the node as a low-dimensional vector. The structure chart is shown in Figure 3.

Given a network G=(V,E) and an adjacency matrix A with nodes before we learn the node embedding representations, we suppose there are n nodes $x_{i}$ in the adjacency matrix A, thus we can define the adjacency matrix A as:(3)$$\;x_{i,j}=\{\begin{array}{c} 1,\;\;x_{i}\;linked\;with\;\;x_{j}\\ 0,\;else \end{array},\;i,\;j=1,2,\;\ldots n$$
(4)$$\;A=\left[\begin{array}{ccc} x_{1,1} & \cdot \cdot \cdot & x_{1,n}\\ \vdots & \ddots & \vdots \\ x_{n,1} & \cdot \cdot \cdot & x_{n,n} \end{array}\right]$$

The second-order proximity is used to indicate the similarity between two neighbor nodes in the network. In particular, the second-order proximity lets nodes with similar neighborhood structures have more similar embedding. Because of the sparsity of networks, it is important that more penalties are imposed on the reconstruction error of the non-zero elements. The second-order loss function is given by:(5)$$L_{2nd}=\sum_{i=1}^{n}\|\left({\hat{x}}_{i}-x_{i}\right)⊙b_{i}\|{}_{2}^{2}=\|\left(\hat{X}-X\right)⊙B{\|}_{F}^{2}$$
where ⊙ indicates the Hadamard product. B is a n×n matrix.$\;b_{ji}=1$, else $b_{i,j}=\beta >1$. $x_{i}$ represents the input vector of ith node and $\hat{x}$ represents the reconstructed vector of the node. For preserving the local network structure, the first-order proximity is regarded as the supervised information to restrain the similarity of unrevealed representations between two nodes. The first-order loss function is given by:(6)$$L_{1st}=\overset{n}{\underset{i,j=1}{\sum }}A_{i,j}\|y_{i}-y_{j}{\|}_{2}^{2}$$

The SDNE loss function combines first-order proximity, second-order proximity and minimizes the following objective function:(7)$$L_{mix}=L_{2nd}+\alpha L_{1st}+\nu L_{reg}$$

Significantly, $L_{reg}$ is a L2−norm regularization term for preventing overfitting. Assume k is the number of hidden layers, $W^{\left(k\right)}$ and ${\hat{W}}^{\left(k\right)}$ are the $k_{th}-layer$ weight matrices and defined as follows:(8)$$Lreg=\frac{1}{2}\overset{k}{\underset{k=1}{\sum }}\left(\|W^{\left(k\right)}{\|}_{F}^{2}+\|{\hat{W}}^{\left(k\right)}{\|}_{F}^{2}\right)$$

Furthermore, SDNE has been adopted to identify lncRNA–protein interactions, lncRNA–disease associations, drug–target interactions and miRNA–disease associations [63,64,65,66]. According to the results of previous studies, SDNE is a high-precision and robust algorithm on a large-scale network. Thus, we employed SDNE to predict underlying DMA in our thesis.

### 2.4. Feature Extraction and Fusion by a Deep Learning Model

CNN and DNN are often used to solve the problem of bioinformatics [67,68]. As shown in Figure 1, CNN is utilized to extract high-level attribute features from the output of word embedding. The CNN operation at layer t can be defined as:(9)$$X_{t}=£(X_{t-1}⊗W_{t}+b_{t})$$
where $W_{t}$ denotes the 4×64 convolution kernel weight matrix, $b_{t}$ the offset vector and $X_{t}$ the attribute feature map. ⊗ represents a convolution operation and £() is the ReLU activation function. To down-sample after convolution operation, we utilized the max-pooling to process the output of the convolution layer. Similarly, we used DNN to further extract topological features from the output of graph embedding. Then, the attribute features and behavior features of miRNAs and drugs are spliced through a concatenate layer. Finally, the output of the concatenate layer is entered into a dense layer. The probability between the miRNA and drug is calculated by a dense layer with softmax activation function. The probability between miRNA and drug can be defined as:(10)$$P=\sigma (f_{ma}⊕f_{mt}⊕f_{da}⊕f_{dt})$$
where $f_{ma}$ is the attribute feature of miRNA, $f_{mt}$ is the topological feature of miRNA, $f_{da}$ is the attribute feature of drug and the $f_{dt}$ is the topological feature of drug. P represents the prediction score, ⊕ denotes the concatenating operation and σ is the softmax activation function. In this model, we selected the Adam algorithm as the optimizer and the binary cross-entropy as the loss function. 

## 3. Results

### 3.1. Performance Evaluation Strategy

To evaluate the performance of the proposed methods, several evaluation metrics were implemented. A five-fold cross-validation was used to verify the proposed method. All of the known samples in each dataset were divided into five subsets in equal measure; the five subsets took turns to serve as the testing set and the other four subsets were used to train the model. Furthermore, the extensively used evaluation criteria were used to evaluate the proposed method, including accuracy (Acc.), sensitivity (Sen.), specificity (Spec.), also precision (Prec.). The Matthews correlation coefficient (MCC) was defined as:(11)$$Acc.=\frac{TN+TP}{TN+TP+FN+FP}$$
(12)$$Sen.=\frac{TP}{FP+FN}$$
(13)$$Spec.=\frac{TN}{TN+FP}$$
(14)$$\mathrm{Pr}ec.=\frac{TP}{TP+FP}$$
(15)$$MCC=\frac{TP\times TN-FP\times FN}{\sqrt{\left(TP+FP\right)\left(TP+FN\right)\left(TN+FP\right)\left(TN+FN\right)}}$$
where TP is the number of positive samples that are predicted correctly; FN is the number of positive samples that are predicted as negative samples; FP is the number of negative samples that are predicted as positive samples; TN is the number of negative samples that are predicted correctly, respectively. To exhibit the performance of the proposed method, the receiver operating characteristic (ROC) curves and precision-recall (PR) curves were drawn. The area under the ROC curve (AUC) and area under the PR (AUPR) curve were calculated as a numerical evaluation of model performance [69,70]. The value of the AUC was generally in the range from 0.5 to 1, where 0.5 denoted a purely random prediction and 1 denoted a perfect prediction. 

### 3.2. Assessment of Prediction Ability

In this section, we evaluated the proposed method under a five-fold cross-validation based on three datasets. Firstly, the known association pairs were regarded as positive samples, and the same number of non-association pairs were chosen randomly as negative samples. The whole dataset was then randomly divided into five parts of the same size. When one subset was used as a test set, the other four subsets were used as training sets to construct features and train the model.

To better evaluate the prediction ability of the proposed methods, we used evaluation indicators such as accuracy (Acc.), sensitivity (Sen.), specificity (Spe.), precision (Prec.) and MCC separately to ensure the comprehensiveness and fairness of the experiment. The results of the five-fold cross-validation on each dataset are shown in Table 2, Table 3 and Table 4. The ROC curves and PR curves of the three datasets can be seen in Figure 3, Figure 4, Figure 5 and Figure 6. Our method achieved average accuracy of 86.46%, 87.56% and 82.16% with standard deviations of 0.48%, 0.30% and 1.14% on the three datasets, respectively (Table 2, Table 3 and Table 4). Figure 4 presents the average AUC and AUPR values of the proposed model on the three datasets. Overall, these results indicate that the MFIDMA model worked well in predictions of DMA.

### 3.3. Comparison with Other Embedding Methods

The topological feature generated by SDNE is an important part of the MFIDMA model. To demonstrate the efficiency of SDNE, we conducted an experiment to compare SDNE and three popular graph embedding methods in different dimensions. Using the same approach of constructing topological features, three other state-of-the-art graph embedding methods (i.e., LINE, Node2vec [71] and Laplacian Eigenmaps (LE) [72]) were utilized to extract the potential graph relationship information and were compared with the SDNE algorithm. The LINE method considered two kinds of proximities, the 1st-order and 2nd-order proximities, and a leverage asynchronous stochastic gradient algorithm (ASGD) [73] to integrate the two kinds of proximities. The Node2vec method is an improved on the DeepWalk method. It uses biased random walks to sample on the network. Laplacian Eigenmaps is a matrix factorization-based method, which can keep two nodes closely embedded when they have high similarity. We set the parameters of these graph embedding methods to their default setting except for the embedding dimension of the output. 

Herein, we discuss the impact of the different embedding dimensions on the model performance, with a range from 32, 64, 128, 256 to 512. We implemented four kinds of graph embedding methods on the RNAInter dataset to obtain the topological features in different dimensions and combined them with the attributed features generated by Word2vec to construct a similar MFIDMA model. The experimental results of all models are illustrated in Figure 7 and Figure 8. The *y*-axis denotes the average AUC values and AUPR values obtained by the corresponding model under the five-fold cross-validation. The *x*-axis denotes five types of embedding dimensions. It is apparent from Figure 7 and Figure 8 that the model using SDNE yielded the best AUC values and AUPR values among the four kinds of embedding methods in the different embedding dimensions. Furthermore, closer inspection of Figure 7 and Figure 8 shows that the prediction model using SDNE achieved the best AUC of 0.9444 and the best AUPR of 0.9382. In conclusion, the SDNE algorithm can learn topological features from a large and sparse network like a drug–miRNA association network better than other graph embedding methods. In addition, the combination of the autoencoder and Laplacian eigenmaps is another reason why the SDNE can effectively extract relationship information from the graph.

Moreover, Figure 7 and Figure 8 show that the best AUC and AUPR are obtained when the embedding dimension is 64. Thus, we set the embedding dimension of SDNE to 64 in this study.

### 3.4. Comparison with Other Classifiers

In this study, we leveraged CNN and DNN to integrate the topological feature and attributed feature and complete the potential DMA prediction task. To discuss the impact of the classifier on the proposed model, we compared our model with different classical classifier in machine learning. It should be noted that we maintained the same feature construction method and changed the classification model. Random forest (RF), Naïve Bayes (NB), support vector machine (SVM) and Logistic Regression (LR) were compared with our model. We employed the grid search method to find the optimal of SVM and RF. There are two parameters of SVM that need to be optimized: c (penalty parameters) and g (kernel function parameters). In the experiments on the SM2miR dataset, we set c to 1 and g to 0.2. We also carried out the grid search method to optimize the three parameters of RF. We set the n_estimator to 100, min_samples_split to 80 and min_samples_leaf to 10. In order to highlight the effect of the classifier model, we chose a relatively small dataset to reduce the influence of features on the classifier. We fed the features generated on the SM2miR dataset into each classifier. The results of the five-fold cross-validation are shown in Table 5. For intuitive comparison, the ROC curve and PR curve of each classification model are shown in Figure 9. As shown in Table 5, RF, NB, SVM and LR obtained average accuracy of 76.88%, 71.95%, 79.81% and 80.40%, respectively. Our model achieved the highest accuracy of 82.16%. Our model also achieved the best results in the ROC curves and PR curves, with AUC values of 0.8944 and AUPR values of 0.8818. Based on the results, the combination of CNN and DNN was an effective method to infer potential DMA. 

### 3.5. Ablation Experiment

To evaluate the role of different features in the proposed method, we explored two types of features. In this study, we constructed an attribute feature, a topological feature and a combined feature to train the computational model on the three datasets. Figure 10, Figure 11 and Figure 12 represent the results of the five-fold cross-validation generated using different models with different features on the three datasets.

Figure 10, Figure 11 and Figure 12 show that the attribute feature performed better than topological features on small datasets. Extracting the attribute feature only required the SMILES of drugs and the sequence of miRNAs. It was difficult to extract information from the association relationships since the limited number of association pairs in the small datasets. The topological feature performed well on datasets that were large and dense. This indicated that the topological features make use of the structural information of the known association network to predict the potential association pairs. The deficiency of the proposed method was the cold start problem. When a new drug or miRNA was added to the network, the prediction performance of the proposed method was not satisfactory because of no known association for reference. The attribute features were more practical for representing new samples. Overall, the gap between the two different features in predicting DMA was limited. These results indicated that we should flexibly combine the two kinds of features according to the scale of the datasets.

### 3.6. Method Comparison Experiment

To further demonstrate the performance of this model, we compared the proposed method with other existing link prediction methods based on the average AUC metric (i.e., Neighbor-based CF, mRNA-based CF, SVD-based MF, EPLMI, MDIPA and GCMDR) [42,74,75,76]. Neighbor-based CF, miRNA-based CF and drug-based CF required self-similarity calculated by the Pearson correlation coefficient of the miRNA and drug and used the collaborative filtering method to infer the potential DMA. The SVD-based MF predicted the DMA by factorizing the adjacency matrix of the miRNAs and drugs. The EPLMI method is a tow-way diffusion model based on the profile similarity, which is proposed to predict the lncRNA and miRNA association. The MDIPA is a novel DMA prediction method based on the self-similarity matrix and neighbor information. The GCMDR is an end-to-end model combining an autoencoder and GCN for predicting DMA. All of the different methods were implemented for the prediction of DMA on the ncDR dataset. The result of the five-fold cross-validation is shown in Table 6. As the result, the MFIDMA model outperformed the second-ranked model with 0.0048 in AUC value. In conclusion, the results indicated that the proposed method with a better performance than previous computational methods could be a reliable computational approach for the prediction of DMA on a large scale.

## 4. Case Study

To further evaluate the prediction capability of the MFIDMA method, we selected the miRNA hsa-let-7c-5p and the drug Verapamil as objects to implement the proposed method as case studies based on the SM2miR v1.0 database. For Verapamil, we removed 167 known DMA related to Verapamil from the dataset; the remaining association were regarded as positive samples. Negative samples were randomly selected from the non-association pairs in the dataset and on the same scale as the positive samples. The combination of the positive samples and negative samples was treated as the training set to train the model. We then connected hsa-let-7c-5pl with the other drugs for validation. After sorting the results of the prediction scores in descending order, 9 of the top 15 candidate drugs were verified by the PubMed literature. The result of the validation is shown in Table 7, and some supporting evidence was found. For example, the expression level of hsa-let-7c-5p reduced in cells resistant to gemcitabine [77]. Through inactivating the IL-6/STAT3 pathway, transfection of hsa-let-7c-5p recovered the sensitivity to cisplatin [78]. The sensitivity of 5-Fluorouracil was influenced by Akt2, which declined due to the over-activating of hsa-let-7c-5p [79]. Fulvestrant regulated the expression of hsa-let-7c-5p to affect Gefitinib [80]. Moreover, the same approach was implemented on Verapamil with 5573 positive samples. Table 8 shows 10 of the top 15 candidate miRNAs that were verified from the RNAInter database, and we have evidence to support them. For example, hsa-miR-34a-5p was down-regulated in Verapamil-resistant MCF-7 breast cancer cells [81]. Hsa-miR-21-5p and hsa-miR-15a-5p played regulatory roles in MCF7/AdrVp [82]. The results of the case studies indicated that the proposed method could predict the drug–miRNA association with high efficiency and robustness. 

## 5. Conclusions

In general, it seems that as the understanding of molecular mechanisms improve, it is suggested that the abnormal expression level of miRNA is associated with diseases. Micro-RNA also offers a new insight into drug-target selection. Discovering DMA is crucial to developing miRNA therapeutics and miRNA-target drugs. Consequently, several studies have investigated the computational model to identify DMA. Herein, our study has offered a multiple feature integrated model, MFIDMA, to identify the potential association between drugs and miRNAs. In MFIDMA, we formulated the drug–miRNA network and utilized SDNE to obtain the topological features. The miRNA sequences and drug SMILES were regarded as a biological sentence and generated attribute features using the Word2vec algorithm. The DNN and CNN models were then used to extract deep learning information. Finally, the predicted results of DMA were obtained using a full connection layer with integrated features. To assess the MFIDMA model, this was implemented on three datasets with a five-fold cross-validation. Our model achieved average AUC values of 0.9407, 0.9444 and 0.8919 on three of the datasets we collected. In addition, we carried out case studies and comparative experiments with other existing methods. Comprehensively, the results of the abovementioned experiments illustrated that the proposed model can predict DMA precisely and robustly. Moreover, in MFIDMA, we used miRNA sequence information and drug SMILES instead of self-similarity, which allowed our model to process new miRNAs and drugs. Future research will attempt to use side information about miRNAs and drugs such as miRNA family information, drug fingerprints and miRNA-gene information.

## Figures and Tables

**Figure 1 biology-12-00041-f001:**
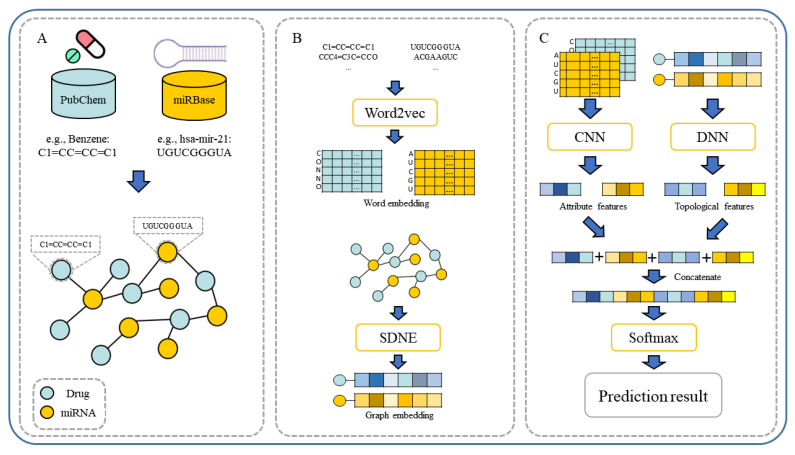
The flowchart of the proposed model. (**A**). Construction of drug–miRNA association network. (**B**). The workflow of word embedding and graph embedding. (**C**). The workflow of feature fusion and prediction.

**Figure 2 biology-12-00041-f002:**
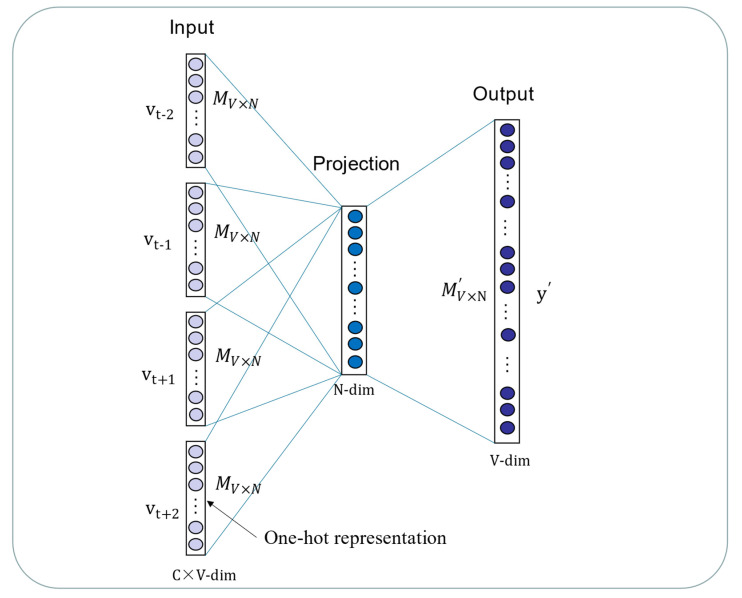
The CBOW model is constructed by the input layer, projection layer and output layer.

**Figure 3 biology-12-00041-f003:**
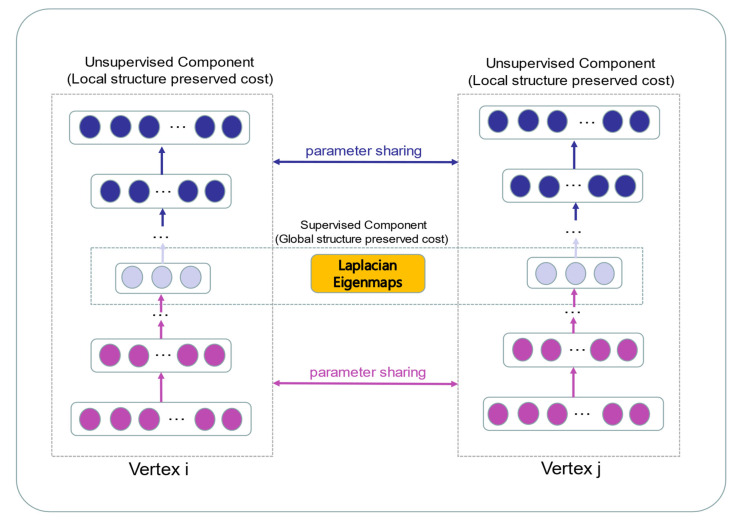
The schematic diagram of the SDNE framework.

**Figure 4 biology-12-00041-f004:**
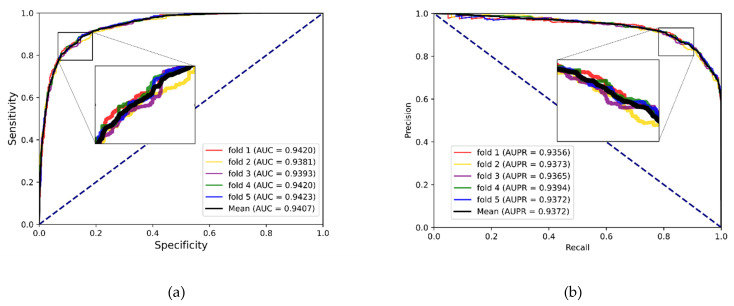
(**a**) The ROC curves of the result on the ncDR datasets. (**b**) The PR curves of the result on the ncDR datasets.

**Figure 5 biology-12-00041-f005:**
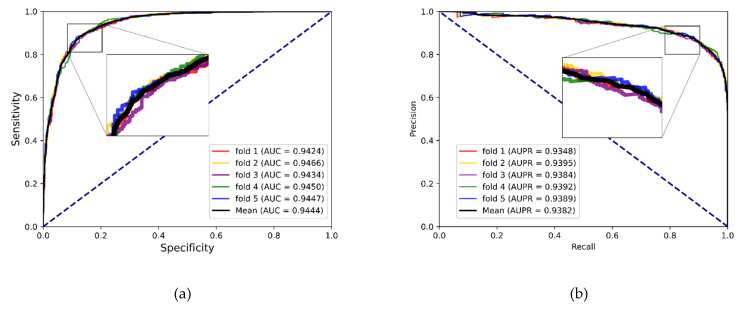
(**a**) The ROC curves of the result on the RNAInter datasets. (**b**) The PR curves of the result on the RNAInter datasets.

**Figure 6 biology-12-00041-f006:**
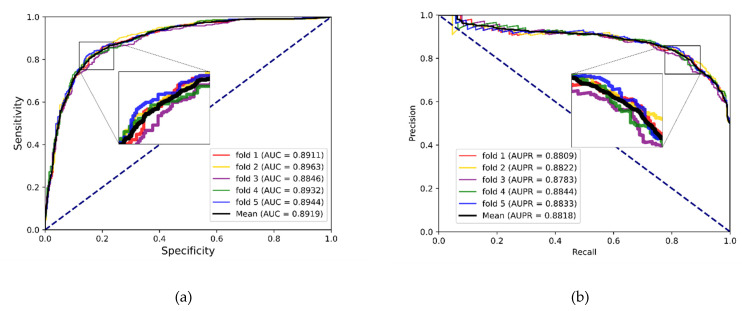
(**a**) The ROC curves of the result on the SM2miR datasets. (**b**) The PR curves of the result on the SM2miR datasets.

**Figure 7 biology-12-00041-f007:**
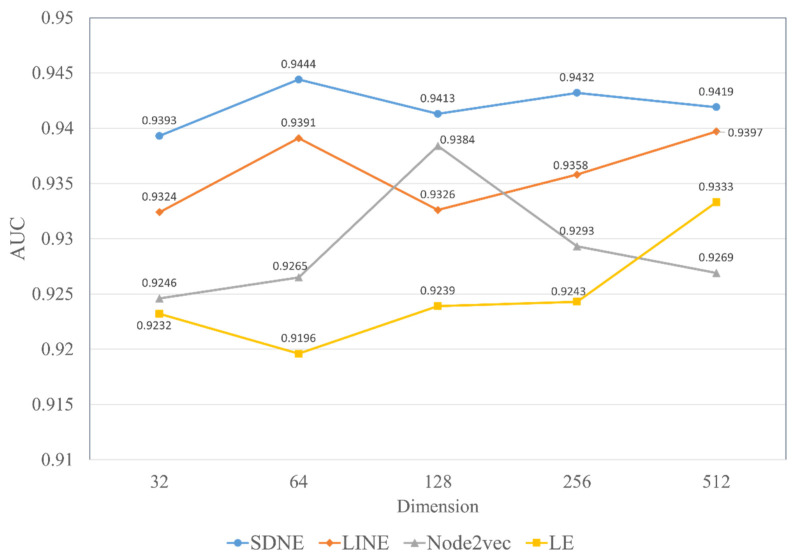
The AUC values of using four network embedding methods in different dimensions.

**Figure 8 biology-12-00041-f008:**
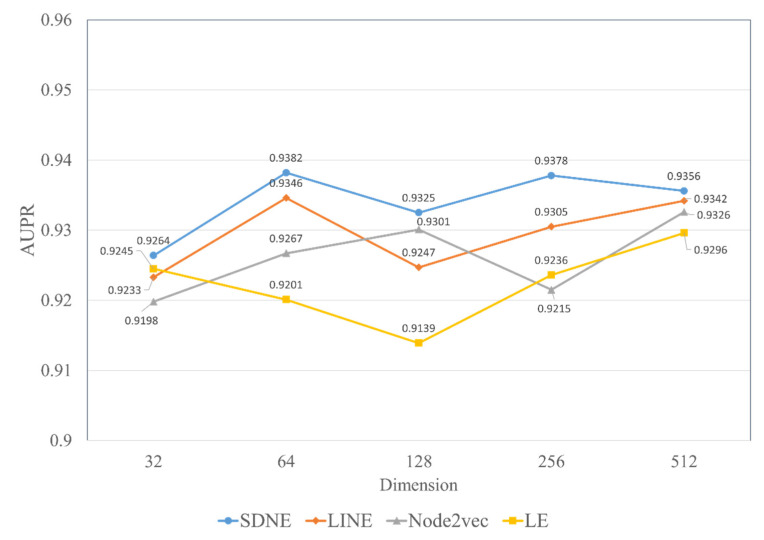
The AUPR values of using four network embedding methods in different dimensions.

**Figure 9 biology-12-00041-f009:**
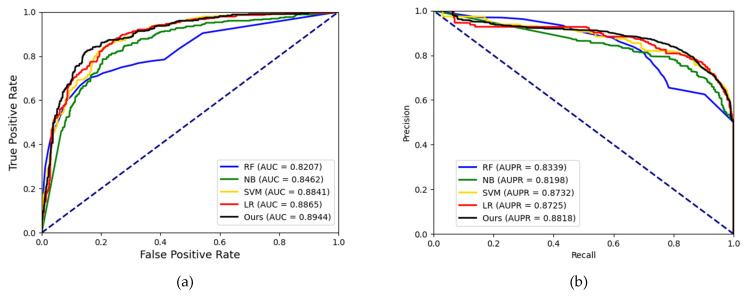
(**a**) The ROC curves of the result generated by different classifiers using the SM2miR dataset. (**b**) The PR curves of the result generated by different classifiers using the SM2miR dataset.

**Figure 10 biology-12-00041-f010:**
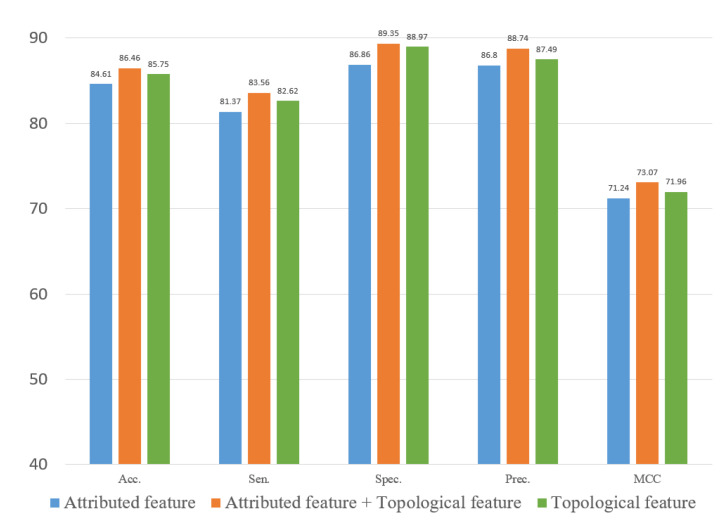
Result of ablation experiment on the ncDR dataset.

**Figure 11 biology-12-00041-f011:**
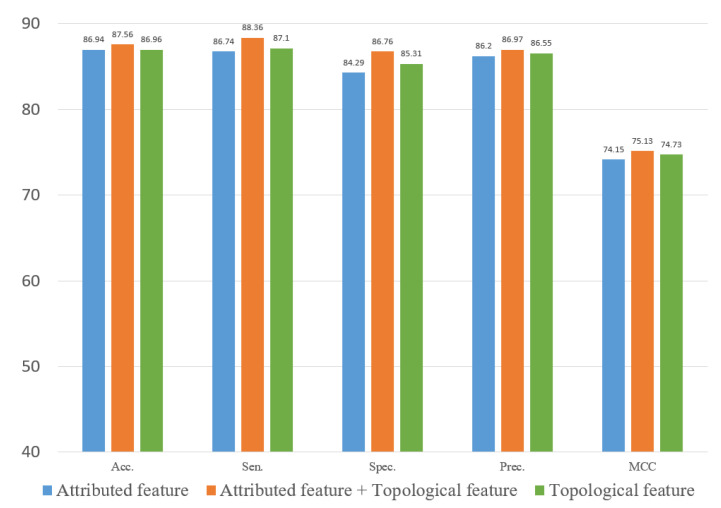
Result of ablation experiment on the RNAInter dataset.

**Figure 12 biology-12-00041-f012:**
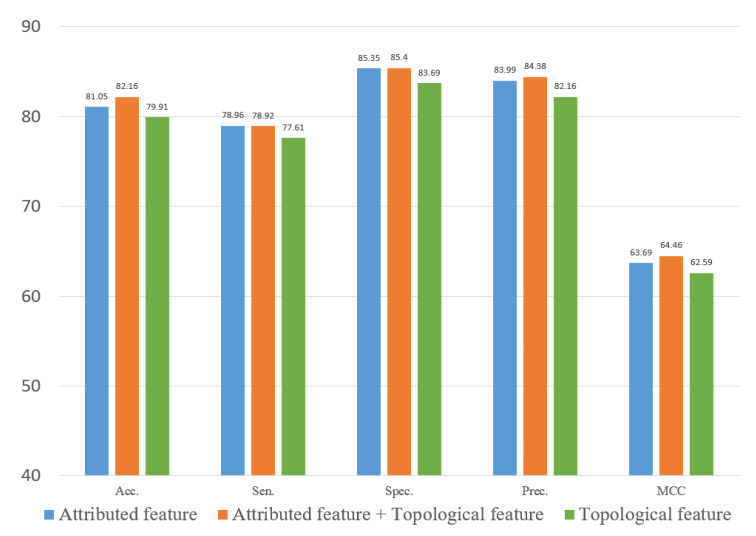
Result of ablation experiment on the SM2miR dataset.

**Table 1 biology-12-00041-t001:** The statistics of miRNAs, SMs and SM–miRNA association in three datasets.

Databases	ncDR	RNAInter	SM2miR
Drug	95	283	138
miRNA	624	1009	580
Associations	4457	5740	2126

**Table 2 biology-12-00041-t002:** The performance of the proposed method in ncDR datasets.

Fold	Acc. (%)	Sen. (%)	Spec. (%)	Prec. (%)	MCC (%)
1	86.94	82.51	91.37	90.53	74.17
2	86.32	84.19	88.45	87.94	72.71
3	85.82	80.94	90.7	89.69	71.98
4	86.94	84.64	89.24	88.72	73.96
5	86.27	85.54	87.00	86.80	72.54
Average	86.46 ± 0.48	83.56 ± 1.83	89.35 ± 1.75	88.74 ± 1.46	73.07 ± 0.95

**Table 3 biology-12-00041-t003:** The performance of the proposed method in RNAInter datasets.

Fold	Acc. (%)	Sen. (%)	Spec. (%)	Prec. (%)	MCC (%)
1	87.20	88.50	85.89	86.25	74.42
2	87.72	88.85	86.59	86.88	75.45
3	87.28	87.11	87.46	87.41	74.56
4	87.81	88.33	87.28	87.41	75.61
5	87.80	89.02	86.59	86.90	75.63
Average	87.56 ± 0.3	88.36 ± 0.75	86.76 ± 0.63	86.97 ± 0.48	75.13 ± 0.59

**Table 4 biology-12-00041-t004:** The performance of the proposed method in SM2miR datasets.

Fold	Acc. (%)	Sen. (%)	Spec. (%)	Prec. (%)	MCC (%)
1	81.34	77.93	84.74	83.63	62.82
2	82.98	80.52	85.45	84.69	66.04
3	80.63	76.53	84.74	83.38	61.48
4	82.51	78387	86.15	85.06	65.20
5	83.33	80.75	85.92	85.15	66.76
Average	82.16 ± 1.14	78.92 ± 1.78	85.40 ± 0.65	84.38 ± 0.82	64.46 ± 2.23

**Table 5 biology-12-00041-t005:** The performance of the proposed method in SM2miR datasets.

	RF	NB	SVM	LR	Ours
Acc. (%)	76.88 ± 2.13	71.95 ± 1.96	79.81 ± 1.29	80.40 ± 1.94	82.16 ± 1.14
Sen. (%)	66.67 ± 1.56	52.35 ± 1.84	76.53 ± 1.94	77.00 ± 1.58	78.92 ± 1.78
Spec. (%)	87.09 ± 0.96	91.55 ± 1.03	83.10 ± 0.73	83.80 ± 0.96	85.40 ± 0.65
Prec. (%)	83.78 ± 1.95	86.10 ± 0.65	81.91 ± 0.96	82.62 ± 0.73	84.38 ± 0.82
MCC (%)	54.91 ± 2.65	47.72 ± 2.61	59.75 ± 2.01	60.94 ± 2.10	64.46 ± 2.23

**Table 6 biology-12-00041-t006:** Comparison of the prediction performance based on ncDR datasets.

Methods	Average AUC
Neighbor-based CF	0.8644 ± 0.0009
Drug-based CF	0.7313 ± 0.0008
miRNA-based CF	0.8235 ± 0.0015
SVD-based CF	0.6007 ± 0.0052
EPLMI	0.8971 ± 0.0009
MDIPA	0.9081 ± 0.0038
GCMDR	0.9359 ± 0.0006
MFIDMA	0.9407 ± 0.0019

**Table 7 biology-12-00041-t007:** The top 15 predicted drugs interacting with the miRNA hsa-let-7c-5p.

Rank	Drug	PubChem ID	miRNA	Evidence
1	Gemcitabine	60750	hsa-let-7c-5p	confirmed
2	5-Fluorouracil	3385	hsa-let-7c-5p	confirmed
3	Cisplatin	2767	hsa-let-7c-5p	confirmed
4	Eloxatine	5310940	hsa-let-7c-5p	confirmed
5	Doxorubicin	31703	hsa-let-7c-5p	confirmed
6	Paclitaxel	36314	hsa-let-7c-5p	unconfirmed
7	Ginsenoside Rh2	119307	hsa-let-7c-5p	confirmed
8	D-Glucose	5793	hsa-let-7c-5p	unconfirmed
9	Sunitinib	5329102	hsa-let-7c-5p	unconfirmed
10	Verapamil	2520	hsa-let-7c-5p	unconfirmed
11	Vincristine	5978	hsa-let-7c-5p	confirmed
12	Tamoxifen	2733526	hsa-let-7c-5p	unconfirmed
13	Gefitinib	123631	hsa-let-7c-5p	confirmed
14	Etoposide	36462	hsa-let-7c-5p	unconfirmed
15	PLX-4720	24180719	hsa-let-7c-5p	confirmed

**Table 8 biology-12-00041-t008:** The top 15 predicted miRNAs interacting with the Verapamil.

Rank	miRNA	Drug	PubChem ID	Evidence
1	hsa-miR-34a-5p	Verapamil	2520	confirmed
2	hsa-miR-16-5p	Verapamil	2520	confirmed
3	hsa-miR-155-5p	Verapamil	2520	confirmed
4	hsa-miR-221-3p	Verapamil	2520	confirmed
5	hsa-miR-21-5p	Verapamil	2520	confirmed
6	hsa-miR-200b-3p	Verapamil	2520	unconfirmed
7	hsa-miR-203a-3p	Verapamil	2520	unconfirmed
8	hsa-miR-500a-5p	Verapamil	2520	unconfirmed
9	hsa-miR-146a-5p	Verapamil	2520	confirmed
10	hsa-miR-24-3p	Verapamil	2520	unconfirmed
11	hsa-miR-145-5p	Verapamil	2520	confirmed
12	hsa-miR-200c-3p	Verapamil	2520	confirmed
13	hsa-miR-629-5p	Verapamil	2520	confirmed
14	hsa-miR-29a-3p	Verapamil	2520	confirmed
15	hsa-miR-126-3p	Verapamil	2520	unconfirmed

## Data Availability

The datasets for this study can be found in the ncDR [http://www.jianglab.cn/ncDR/index.jsp], RNAInter [www.rnainter.org] and SM2miR [http://www.jianglab.cn/SM2miR]. The data and source code can be found at https://github.com/Heath0/MFIDMA/tree/master.

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
