# Peer review of "MFIDMA: A Multiple Information Integration Model for the Prediction of Drug–miRNA Associations"

_biology, 2022, doi:10.3390/biology12010041_

Round 1
Reviewer 1 Report
Identification of drug-miRNA association is an important and interesting topic in biology and bioinformatics. In this manuscript, a bipartite graph was constructed and structural deep network embedding was used to obtain topological features. In addition, word2vec algorithm was also utilized to calculate attribute features of miRNA and drug. Finally, convolution neural network and deep neural network were employed to integrate topological features and attribute features, and identify potential associations between drug and miRNA. Performance of the current method was evaluated and validated by using 5-fold cross-validation and case study. However, some important issues should be suitably addressed before its publication. My detailed comments are as follows:
1. Line 48, fu·nction?
2. In figure 1. How to integrate features and classify? After two kinds of features were input into convolutional neural network and deep neural network, respectively. Detailed description is not included in the manuscript.
3. Line 135, five datasets?
4. Line 152 and 153, the RNAInter, ncDR and SM2miR datasets include altogether 10571 drug-miRNA associations. Why are there 12323 drug-miRNA associations after only human related associations are selected?
5. Line 159, 518 drugs and 2213 miRNAs. However, 492 drugs and 1788 miRNAs have SMILES and sequences. How to deal with those drugs without SMILES and miRNAs without sequences?
6. The title and content of Table 1 are incorrect.
7. Line 208 and 209, each letter in sequences will be represented as a vector with dimension 64. Because the SMILES lengths of drug are different, the length of feature are also different. How to solve this problem?
8. Line 230, represent the node as a low-dimensional vector. What is the dimension?
9. Line 264 and 265, Acc., Spec. and Prec. What does “.” mean?
10. Equation 11, “Sen.”, however, “Sen” in lien 264.
11. Line 277, 5 datasets?
12. Table 3 and Table 4 are not described in this manuscript.
13. Results listed in line 286, 287 and 288 can’t be seen in the Table 2.
14. Line 290, five data sets?
15. Table 3, “+”?
16. Table 2, 86.46±0.48, 0.48 is the standard deviation? Or relative standard deviation?
17. 3.4 Compare with other classifiers, the mode parameters of random forest and support vector machine should be optimized. Otherwise, this comparison is unfair.
18. The current method should be compared with the latest literature methods to further demonstrate the performance.
19. Line 379, 3.6 ablation experiment?
20. Line 408, Table 4?
21. Line 415, Table 5?
22. Line 438, data0333?
23. Line 439, 0sets?
24. The detailed architectures and optimizers of convolutional neural network and deep neural network should be described.
25. Line 461, https://github.com/Heath0/MFIDMA is invalid.
Author Response
Response to Reviewer 1 Comments
Point 1: Line 48, fu·nction?
Response 1: Thanks for your comment, we have revised this in line 57 and checked the rest of the article.
Point 2: In figure 1. How to integrate features and classify? After two kinds of features were input into convolutional neural network and deep neural network, respectively. Detailed description is not included in the manuscript.
Response 2:Thank you for your valuable suggestion. We have redrawn Figure 1 and updated the figure caption for more detail. The way to integrate features is concatenate attributed features and topological features of miRNA and drug. And the integrated features are input into the final dense layer as sample features to calculate the probability of the association between miRNA and drug. Relevant description are described in detail in section 2.4.
Point 3: Line 135, five datasets?
Response 3: Thanks for your comment, we have corrected this part in line 145.
Point 4:Line 152 and 153, the RNAInter, ncDR and SM2miR datasets include altogether 10571 drug-miRNA associations. Why are there 12323 drug-miRNA associations after only human related associations are selected?
Response 4: Thank you for pointing out this problem in the manuscript and giving valuable comments. We recounted the sample of our dataset. Before data pre-processing, we collected a total of 19310 miRNA drug interaction samples from ncDR, RNAInter and SM2miR dataset websites. Because the dataset has been adjusted before and after the experiment, the wrong description of data appears in the text. We have updated the data in the article in 2.1 section.
Point 5: Line 159, 518 drugs and 2213 miRNAs. However, 492 drugs and 1788 miRNAs have SMILES and sequences. How to deal with those drugs without SMILES and miRNAs without sequences?
Response 5: Thanks for your comment. Due to the update of the dataset website, some unique codes of miRNAs or drug may become invalid. This is an important reason for missing information, so we will eliminate the sample pairs with missing information in data pre-processing. After re counting, we collected different 470 types of drugs and 1623 different types of miRNAs. We collect superfluous drug SMILES and miRNA sequences from PubChem and miRBase, which can be used as a corpus for word embedding.
Point 6: The title and content of Table 1 are incorrect.
Response 6: Thanks for your comment, we have corrected the Table 1.
Point 7: Line 208 and 209, each letter in sequences will be represented as a vector with dimension 64. Because the SMILES lengths of drug are different, the length of feature are also different. How to solve this problem?
Response 7: Thanks for your comment. Because the length of most miRNAs and drugs will not exceed 25 and 64. For the miRNA sequences and drug SMILES, we set the maximum intercepted length to 25 and 64 respectively, and add 0 to the insufficient vector.
Point 8: Line 230, represent the node as a low-dimensional vector. What is the dimension?
Response 8: Thanks for your comment. We set the output dimension of SDNE to 64, therefore each node can be represented as a 64-dimensional vector.
Point 9: Line 264 and 265, Acc., Spec. and Prec. What does “.” mean?
Response 9: Thanks for your comment, “.” means that the term of evaluation criteria is represented in an abbreviated form.
Point 10: Equation 11, “Sen.”, however, “Sen” in lien 264.
Response 10: Thank you for pointing out this problem. We have corrected “Sen” to “Sen.”.
Point 11: Line 277, 5 datasets?
Response 11: Thanks for your comment, we have corrected this in line 319.
Point 12: Table 3 and Table 4 are not described in this manuscript.
Response 12: Thanks for your comment,We have mentioned in line 327.
Point 13: Results listed in line 286, 287 and 288 can’t be seen in the Table 2.
Response 13: We are grateful for the suggestion. We have mentioned in line 329.
Point 14: Line 290, five data sets?
Response 14: Thanks for your comment,We have corrected in line 333.
Point 15: Table 3, “+”?
Response 15: Thanks for your comment,We have corrected “+” to “±” in Table 3.
Point 16: Table 2, 86.46±0.48, 0.48 is the standard deviation? Or relative standard deviation?
Response 16: Thanks for your comment. In this paper, the number after “±” is the standard deviation. The relevant content is mentioned in lines 330.
Point 17: 3.4 Compare with other classifiers, the mode parameters of random forest and support vector machine should be optimized. Otherwise, this comparison is unfair.
Response 17: We gratefully appreciate for your valuable suggestion. As suggested by the reviewer, we have optimized the parameters of SVM and RF using grid search and added parameter settings in the section 3.4.
Point 18: The current method should be compared with the latest literature methods to further demonstrate the performance.
Response 18: Thanks for your suggestion. We compared the current method with MDIPA [1], EPLMI[2] and GCMDR[3] in 3.6 section.
[1]Jamali, A.A.; Kusalik, A.; Wu, F.-X.J.B. MDIPA: a microRNA–drug interaction prediction approach based on non-negative matrix factorization. 2020, 36, 5061-5067.
[2]Huang, Y.-A.; Chan, K.C.; You, Z.-H.J.B. Constructing prediction models from expression profiles for large scale lncRNA–miRNA interaction profiling. 2018, 34, 812-819.
[3]Huang, Y.-a.; Hu, P.; Chan, K.C.; You, Z.-H.J.B. Graph convolution for predicting associations between miRNA and drug resistance. 2020, 36, 851-858.
Point 19: Line 379, 3.6 ablation experiment?
Response 19: We appreciate very much the reviewer’s comments. We have corrected the title of section 3.6 to “Method Comparison Experiment”.
Point 20: Line 408, Table 4?
Response 20: Thanks for your comment, we have corrected the number of the Table in line 487.
Point 21: Line 415, Table 5?
Response 21: Thanks for your comment, we have corrected the number of the Table in line 488.
Point 22: Line 438, data0333?
Response 22: Thanks for your comment, we have corrected the incorrect line break in line 505.
Point 23: Line 439, 0sets?
Response 23: Thanks for your comment, we have corrected the incorrect line break in line 505.
Point 24: The detailed architectures and optimizers of convolutional neural network and deep neural network should be described.
Response 23: Thank you for your very helpful and constructive comments. We add a section 2.4 “Feature extraction and fusion by a deep learning model” to describe architectures and optimizers of convolutional neural network and deep neural network in detail.
Point 25: Line 461, https://github.com/Heath0/MFIDMA is invalid.
Response 25: Thank you for pointing out this problem. We have uploaded the code again and attached the address https://github.com/Heath0/MFIDMA/tree/master at the end of the paper. Thank you for your careful review. We really appreciate your efforts in reviewing our manuscript during this unprecedented and challenging time. We wish good health to you, your family, and community. Your careful review has helped to make our study clearer and more comprehensive.

Reviewer 2 Report
The authors proposed a new method for predicting small drug-miRNA association based on word2vec and graph embedding methods. The results of cross-validation show that the method has good performance. Identifying the regulatory relation between miRNA and drug is very important. The topic is creative and meaningful and the design of the constructed is reasonable. The manuscript could be accepted after solving the following suggestions.
1. The clarity of Fig. 1 is not enough, the authors are supposed to further improve the clarity of it.
2. What is the statistical significance of the results after 5-fold validation?
3. In the ablation experiment, the other evaluation criteria of results should also be listed.
4. Your manuscript needs careful editing and particular attention to English grammar, spelling, and sentence structure. For example, there is an incorrect line break in line 438.
Author Response
Response to Reviewer 2 Comments
Point 1: The clarity of Fig. 1 is not enough, the authors are supposed to further improve the clarity of it.
Response 1: Thank you very much for the suggestion. We would provide a clearer and more explanatory Figure 1.
Point 2: What is the statistical significance of the results after 5-fold validation?
Response 2: Cross-Validation is a widely used method of evaluating and comparing learning algorithms by dividing data into two segments: one used to learn or train a model and the other used to validate the model. In typical cross-validation, the training and validation sets must cross-over in successive rounds such that each data point has a chance of being validated against.
In k-fold cross-validation, the original sample is randomly partitioned into k equal sized subsamples. Of the k subsamples, a single subsample is retained as the validation data for testing the model, and the remaining k-1 subsamples are used as training data. The cross-validation process is then repeated k times, with each of the k subsamples used exactly once as the validation data. The k results from the folds can then be averaged to produce a single estimation.
Point 3: In the ablation experiment, the other evaluation criteria of results should also be listed.
Response 3: Thanks for your suggestion. We added other evaluation criteria of the ablation results in Figure 10-12.
Point 4: Your manuscript needs careful editing and particular attention to English grammar, spelling, and sentence structure. For example, there is an incorrect line break in line 438
Response 4: In addition, we have asked several colleagues who are skilled authors of English language papers to check the English. We believe that the language is now acceptable for the review process.
Thank you for your careful review. We really appreciate your efforts in reviewing our manuscript during this unprecedented and challenging time. We wish good health to you, your family, and community. Your careful review has helped to make our study clearer and more comprehensive.

Reviewer 3 Report
1. There exist some problems with the grammar and format of this manuscript, such as ‘low relative’ on line 36, ‘lack is known’ on line 83, Table 1 at line 174 and error on lines 438 to 439.
2. The description of the data set is not clear, nor is how to construct a bipartite graph from sequence data, and it is necessary to explain these two questions for the integrity of the article.
3. The figure 2 shows how to represent miRNAs and drugs, but the explanation of figure 2 is relatively short, and it needs more description of “MV×M” and “M’ V×M” in particular.
4. The formulas 3,4 and 5 are incorrect, correct and explain them.
5. The code in the manuscript is not available, you can re-upload it to github and replace the original unavailable address with the new one.
6. The format of references does not meet the requirements of the journal. E.g., the description of LINE algorithm first appears on line 223 but is not referenced until line 308. And the Neighbor-based CF, mRNA-based CF, SVD-based MF algorithms are not referenced in the manuscript at line 382.
7. Accuracy, sensitivity, specificity, precision, and MCC are mentioned in section 3.1 of the article, and the values of these metrics in the three datasets are obtained. However, only the SM2miR dataset is used to compare these metrics in 3.4, while the other two datasets do not use them, please explain reasons.
Author Response
Response to Reviewer 3 Comments
Point 1: There exist some problems with the grammar and format of this manuscript, such as ‘low relative’ on line 36, ‘lack is known’ on line 83, Table 1 at line 174 and error on lines 438 to 439.
Response 1: Thanks for your comment, we have corrected these problems. Moreover, we have revised this and checked the rest of the article.
Point 2: The description of the data set is not clear, nor is how to construct a bipartite graph from sequence data, and it is necessary to explain these two questions for the integrity of the article.
Response 2: Thank you for pointing out this problem in the manuscript. In section 2.1, I expanded the description of the dataset and added sentences follow as: “These positive samples can be represented as adjacent edge list and then turn them into a drug-miRNA association bipartite graph. And the miRNA name and sequence recording in miRBase represent the information of each miRNA nodes. Similarly, the information of drug is uniquely identified by the CID and SMILES from PubChem.”
Point 3: The figure 2 shows how to represent miRNAs and drugs, but the explanation of figure 2 is relatively short, and it needs more description of “MV×M” and “M’ V×M” in particular.
Response 3: Thank you for your very helpful and constructive comments. The weights matrix between the input layer and the projection layer is represented by a V×N matrix M. And the M' is not the transpose of M, but a V×N weights matrix between the projection layer and the output layer. For the extended description of Figure 2, I have added it to the line 213.
Point 4: The formulas 3,4 and 5 are incorrect, correct and explain them.
Response 4: Thank you for your very helpful and constructive comments. We have removed redundant formulas and corrected the letters in formulas. Formulas 3 and 4 indicate that we given an adjacency matrix A with n nodes xi, which may facilitate the calculation of graph structure data.
Point 5: The code in the manuscript is not available, you can re-upload it to github and replace the original unavailable address with the new one.
Response 5: Thank you for pointing out this problem. We have uploaded the code again and attached the address https://github.com/Heath0/MFIDMA/tree/master at the end of the paper.
Point 6: The format of references does not meet the requirements of the journal. E.g., the description of LINE algorithm first appears on line 223 but is not referenced until line 308. And the Neighbor-based CF, mRNA-based CF, SVD-based MF algorithms are not referenced in the manuscript at line 382.
Response 6: We are grateful for the suggestion. Relevant articles have been referenced in line 443 and the reference format has been checked.
Point 7: Accuracy, sensitivity, specificity, precision, and MCC are mentioned in section 3.1 of the article, and the values of these metrics in the three datasets are obtained. However, only the SM2miR dataset is used to compare these metrics in 3.4, while the other two datasets do not use them, please explain reasons.
Response 7: Thank you for your common.SM2miR dataset is a relatively small dataset among the three datasets. And the size of the dataset will affect the effect of the graph embedding method. The contribution of behavior features to the prediction model is relatively small, therefore we think this dataset can best reflect the performance of classifiers. The reason for using only this dataset has been added in line 397.
Thank you for your careful review. We really appreciate your efforts in reviewing our manuscript during this unprecedented and challenging time. We wish good health to you, your family, and community. Your careful review has helped to make our study clearer and more comprehensive.

Round 2
Reviewer 1 Report
This manuscript can be published
Reviewer 3 Report
Although you have explained the weight matrix in response 3, the dimension of M’ should be N×V.